# A Matrix Approach for Analyzing Signal Flow Graph

**Shyr-Long Jeng [1],\*, Rohit Roy [2] and Wei-Hua Chieng [2]**

[1]    Department of Mechanical Engineering, Lunghwa University of Science and Technology,
      Taoyuan City 333326, Taiwan

[2]    Department of Mechanical Engineering, National Chiao Tung University, Hsinchu 30010, Taiwan;
      rohitroy41@live.com (R.R.); whc@cc.nctu.edu.tw (W.-H.C.)

\*    Correspondence: aetsl@gm.lhu.edu.tw

**Abstract:** Mason's gain formula can grow factorially because of growth in the enumeration of paths in a directed graph. Each of the $(n-2)!$ permutation of the intermediate vertices includes a path between input and output nodes. This paper presents a novel method for analyzing the loop gain of a signal flow graph based on the transform matrix approach. This approach only requires matrix determinant operations to determine the transfer function with complexity $O(n^3)$ in the worst case, therefore rendering it more efficient than Mason's gain formula. We derive the transfer function of the signal flow graph to the ratio of different cofactor matrices of the augmented matrix. By using the cofactor expansion, we then obtain a correspondence between the topological operation of deleting a vertex from a signal flow graph and the algebraic operation of eliminating a variable from the set of equations. A set of loops sharing the same backward edges, referred to as a loop group, is used to simplify the loop enumeration. Two examples of feedback networks demonstrate the intuitive approach to obtain the transfer function for both numerical and computer-aided symbolic analysis, which yields the same results as Mason's gain formula. The transfer matrix offers an excellent physical insight, because it enables visualization of the signal flow.

**Keywords:** signal flow graph; transfer function; Mason's graph; linear system

## 1. Introduction

A signal flow graph set up directly after inspecting a physical system without first formulating the associated equations is one of the most common tools for representing a complicated linear control system. It offers a visual structure upon which causal relationships among several variables can be compared. In the past several decades, flow graph analysis has been widely used in electrical engineering [1–6], computer science, biological science [7], and for solving economic problems. Furthermore, the applications of graph theory in conjunction with symbolic network analysis [8,9] and the computer-aided simulation of electronic circuits [10] have been widely encountered in recent years.

Mason's gain formula [11,12], or Mason's rule, is a systematic method for obtaining the transfer function of a signal flow graph between input and output nodes, especially for complex and high-dimensional systems. Mason evaluated the determinant of a signal flow graph and proved the rule by considering the determinant value. The advantage of Mason's rule is that it can be drawn directly from the physical system without setting up the equations in matrix form or requiring any reduction procedure for the flow graph [13–15]. Coates described an alternative representation of the flow graph [16,17] that is derived from algebraic equations written in terms of the incidence and weight matrices of the graph. Coates' gain formula can be used to find the transfer function algebraically by labeling each signal, writing the equation for how that signal depends on other signals, and then solving multiple equations for the output signal in terms of the input signal.

Mason's and Coates' gain formulas are classics in flow graph theory [18]. These two formulations are closely related in terms of both the manipulations and topological formulas. Mason's rule not only retains the intuitive character of the block diagram, but also enables determination of the gain between input node and output nodes of a signal flow graph through inspection. Mason's graph is a more natural representation of a physical system than Coates graph. Coates' gain formula, by contrast, computes the output directly regardless of the number of inputs present in the system. Mason's graph reduction rules cannot be applied directly in Coates' gain formula because the direct computation of a given output used in this method is not focused on the same cause and effect formulation of equations as in the case of Mason's rule [18]. Mason's rule involves dividing a signal flow graph into several independent loop gains and analyzing the input–output transfer function. The existence of non-touching loops increases the complexity of the formula. Neither determining the exact number of independent loop gains nor recognizing the touching loops in a complex system is easy. In general, Mason's gain formula is complicated to implement without making mistakes.

This paper proposes a systematic method called the transfer matrix method [19] to determine the transfer function of a system, and it presents a physical insight into the transfer matrix. As an alternative to Coates' gain formula for solving the system, a solution can be obtained by considering the eigenvector of the transfer matrix for a signal flow graph. A recursive reduction of a signal flow graph is performed using cofactor expansion, which successively eliminates nodes to obtain a subgraph layer analysis and merges the results obtained on the submatrices. Each cofactor expansion reduces the order of the associated matrix by one. The system is then separated into several loop groups including backward and forward paths. From the viewpoint of loop groups, the new method provides an excellent physical insight through visualization of the signal flow.

The organization of the report is as follows. First, in Section 2, the theoretical foundations of the transfer matrix and its representation in the signal flow graphs are described. In Section 3, it is explained that the transfer function overcomes the difficulty of implementing the traditional Mason's gain formula. The augment matrix is performed recursively through cofactor expansion to systematically obtain all possible non-touching loop combinations and represent them compactly. The pseudocode has been constructed to determine the transfer function and calculate the loop group gain. In Section 4, a graph decomposition method is introduced to calculate the forward path gain between two nodes with feedback layers. The method in graph decomposition can give an excellent visualization to the signal flow. Compared with the traditional Mason's gain formula, two examples have been presented to illustrate the methods of calculating the loop group gains. The fact that determinant of the cofactor matrix is the same as the loop group gain is discussed in Section 5. When a virtual backward edge from the output node to the input node is added in Section 6, the numerator of the transfer function can be regarded as the forward path gain. The next section discusses the complexity analysis of the algorithm. Finally, Section 8 concludes the report.

## 2. Transfer Matrix Method

As shown in Figure 1, a node-ordered signal flow graph that contains the set of nodes **X** and the directed edge set **E** can be related by the associated equation [19]

$$\mathbf{A}^{\mathbf{T}} \cdot \mathbf{X} = \mathbf{X} \tag{1}$$

where

$$\mathbf{A}^{\mathbf{T}} = \begin{bmatrix} 1 & 0 & \cdots & 0 & 0 \\ a_{0,1} & a_{1,1} & \cdots & a_{n-1,1} & a_{n,1} \\ \vdots & \vdots & \vdots & \vdots & \vdots \\ a_{0,n-1} & a_{1,n-1} & \cdots & a_{n-1,n-1} & a_{n,n-1} \\ a_{0,n} & a_{1,n} & \cdots & a_{n-1,n} & a_{n,n} \end{bmatrix}$$

$$\mathbf{X} = \begin{bmatrix} x_0 & x_1 & \cdots & x_{n-1} & x_n \end{bmatrix}^{T}$$

and $a_{i,j}$ denotes the directed edge (or gain) from source node $x_i$ to sink node $x_j$.

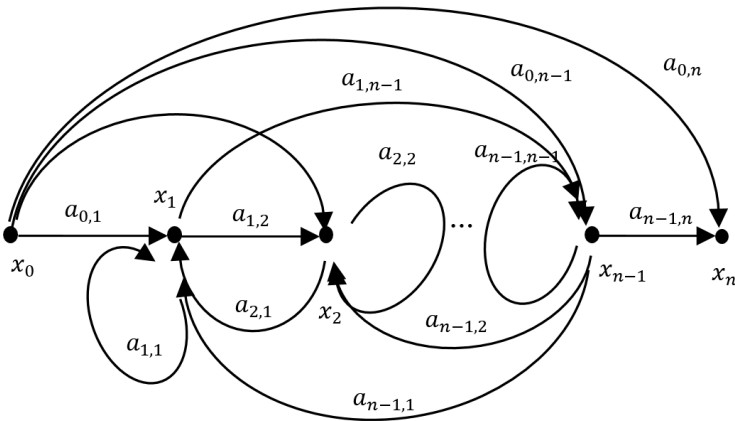

**Figure 1.** A signal flow graph.

The nodes are arranged in a descending order. The edges may be classified as forward, backward, and self-loop edges. When the elements are located below the main diagonal, the associated edges pointing from a source node $x_i$ to sink node $x_j$ satisfy the node ordering, namely that node $x_i$ is numbered prior to node $x_j$. These edges are called forward edges. Against the edge direction, the elements located above the main diagonal are called backward edges. The elements on the diagonal are self-loop edges that consist of only one node. Because the self-loop edge is also one type of backward edge, the elements that are located above the diagonal are called strictly backward edges.

To evaluate the eigenvector of the square matrix $\mathbf{A^T}$ associated with $\lambda = 1$, the first row of $(\mathbf{I} - \mathbf{A^T})$ may be replaced with a row vector $\begin{bmatrix} \hat{x}_0 & \hat{x}_1 & \dots & \hat{x}_n \end{bmatrix}$ that contains symbols for all terms in the numerators of node variables $x_0, x_1, \dots, x_n$, respectively. The augmented matrix $\mathbf{A_a}$ yields

$$\mathbf{A_a} = \begin{bmatrix} \hat{x}_0 & \hat{x}_1 & \dots & \hat{x}_{n-1} & \hat{x}_n \\ -a_{0,1} & (1-a_{1,1}) & \dots & -a_{n-1,1} & -a_{n,1} \\ \vdots & \vdots & \vdots & \vdots & \vdots \\ -a_{0,(n-1)} & -a_{1,(n-1)} & \dots & (1-a_{n-1,n-1}) & -a_{n,n-1} \\ -a_{0,n} & -a_{1,n} & \dots & -a_{n-1,n} & (1-a_{n,n}) \end{bmatrix} \tag{2}$$

The determinant of the augmented matrix $\mathbf{A_a}$ is expressed as the sum of the cofactor of the first row of the matrix multiplied by the corresponding entry in the first row. In other words,

$$\det(\mathbf{A_a}) = \alpha_0 \hat{x}_0 + \alpha_1 \hat{x}_1 + \dots + \alpha_{n-1} \hat{x}_{n-1} + \alpha_n \hat{x}_n \tag{3}$$

The cofactor $\alpha_j$ equals $(-1)^j \det(\mathbf{M_{1,j+1}(A_a)})$, where $(-1)^j \mathbf{M_{1,j+1}(A_a)}$ is the cofactor matrix, a signed version of a minor $\mathbf{M_{1,j+1}(A_a)}$ defined by deleting the first row and the $j$+1th column from the augmented matrix $\mathbf{A_a}$. The ratio between the cofactor $\alpha_j$ attributable to the cofactor $\alpha_0$ satisfies the equilibrium relation between the output node $x_j$ and the input node $x_0$. For a single input multiple output (SIMO) system, the transfer function $G_j$ is then expressed as follows.

$$G_j = \frac{\alpha_j}{\alpha_0} = \frac{(-1)^j \det(\mathbf{M_{1,j+1}(A_a)})}{\det(\mathbf{M_{1,1}(A_a)})} \quad \text{for } j = 1, 2, \dots, n \tag{4}$$

## 3. Mason's Gain Formula

Mason's gain formula [11,12] is a method for finding the transfer function that associates the input and output of a linear signal-flow graph with many variables and loops. The Mason's gain formula is

$$G = \frac{y_{out}}{y_{in}} = \frac{\sum_k G_K \Delta_k}{\Delta}$$

$$\Delta = 1 - \sum L_i + \sum L_i L_j - \sum L_i L_j L_k + \cdots + (-1)^m \sum \cdots$$

where

$y_{in}$ is the input-node variable

$y_{out}$ is the output-node variable

$G$ is the transfer function between $y_{in}$ and $y_{out}$

$\Delta$ is the determinant of the graph

$G_K$ is path gain of the $k$th forward path between $y_{in}$ and $y_{out}$

$\Delta_K$ the cofactor value of $\Delta$ for the $k$th forward path, with the loops touching the $k$th forward path removed

$L_i$ is loop gain of each closed loop in the system

$L_i L_j$ is product of the loop gains of any two non-touching loops

$L_i L_j L_k$ is product of the loop gains of any three pairwise non-touching loops

Consider a signal flow graph with nested loops. The implement procedures of the Mason's gain formula first list all forward paths with corresponding gains and all loops. Once all loops have been generated, they should be combined in all possible ways so that all loops in a combination are non-touching. Make a list of all pairs of non-touching loops taken two, three, four, etc. at a time until no more contact, and multiply their gains ($L_i L_j$, $L_i L_j L_k$, …). Calculate the determinant $\Delta$ and the cofactor $\Delta_K$, and then apply the formula. Lu et al. [20] proposed an algorithm combined with Johnson method for generating the combinations of the non-touching loops. If there are too many variables in the signal flow graph, this method will make the expression of the transfer function very complicated and difficult to analyze. Prasad [15] used tree structure and/or factoring technique to generate and represent non-touching combinations of paths and loops. The tree structure is suitable for the small graph size and most loops are touching. When most loops are non-touching, the factoring method is better to remove combinations of touching loops from all combinations of loops. Beillahi et al. [21] proposed a higher-order logic formation of signal flow graph in HOL Light theorem prover. The touching loop is detected by pre-checking each loop with a higher rank loop than the loop considered in the given loops list. However, these methods are not to generate and represent non-touching combinations of paths and loops systematically and efficiently.

We reduce the signal flow graph recursively through cofactor expansion to systematically obtain all possible non-touching loop combinations and represent them compactly. We also use a set of loops that share the same backward edges, referred to as a loop group, to simplify the loop enumeration. The determinant of the cofactor matrix is equivalent to the loop group gain associated with the backward edges. It provides an excellent physical insight of the signal flow. Although the numerator and denominator of the Mason's gain formula have different forms, our method shows that the numerator maybe regarded as the loop group gain when the unit virtual backward edge from output node to input node is added. The transfer function $G_j$ of the internal node $x_j$ with respect to input node $x_0$ for the SIMO system may be directionally calculated by using matrix determinant according to Equation (4). This method helps to improve clarity and understand the interrelations between entire system and subsystems. Our method is simple and useful enough to teach students the subject of Mason's formula as a part of courses in control systems, digital signal processing, graph theory, and applications.

The pseudocode showing Algorithm 1 is implemented using the transfer matrix method. The pseudocode calculates the transfer function as well as the loop group gain. It takes as an input, the

system matrix $\mathbf{A}^{\mathbf{T}}$ determined in Equation (1) and produces the output as per the requirements. The augmented matrix $\mathbf{A_a}$ is calculated using the operation $(\mathbf{I} - \mathbf{A}^{\mathbf{T}})$.

The transfer function method consists of a numerator and a denominator term. The numerator term is $(-1)^j \det(\mathbf{M_{1,j+1}}(\mathbf{A_a})$ where $j$ represents the output node index and $\mathbf{M_{1,j+1}}(\mathbf{A_a})$ is determined by deleting the first row and $j+1$ column from $\mathbf{A_a}$. The denominator term on the other hand is determined by $\det(\mathbf{M_{1,1}}(\mathbf{A_a}))$, where $\mathbf{M_{1,1}}(\mathbf{A_a})$ is obtained by deleting the first row and the first column of the $A_a$ matrix.

The loop group gain between $p_i$ to $p_k$ is calculated by multiplying the forward path gain from $p_i$ to $p_k$ by the backward edge gain $a_{k,\ i}$. The forward path gain is determined by $\det(^{(j)}H(p_1, \cdots, p_j)|_{(n,n)}^{(m,m+j)})$ as discussed in Section 4, where the arguments from $p_1$ to $p_j$ in descending order denote that the deleted columns are associated with the nodes from $x_{p_1}$ to $x_{p_j}$, respectively.

---

**Algorithm 1. Transfer matrix method's workflow**

---

**Pseudocode:**

  **Requirement:**

    (1) transfer function $G_j$ with output node $p_j$ and input node $p_0$

    (2) loop group gain with nodes from $p_i$ to $p_k$

  **Input:**

    adjacent metric $\mathbf{A}^{\mathbf{T}}$

    If Requirement == transfer function:

        Input.append(0, j);

    else if Requirement == loop group gain:

        Input.append(i, k);

  **Output:** required result

    augmented matrix $\mathbf{A_a} \leftarrow (\mathbf{A}^{\mathbf{T}} - \mathbf{I})$

    If Requirement == Transfer function:

        $\alpha_j \leftarrow (-1)^j \det(\mathbf{M_{1,j+1}}(\mathbf{A_a}))$

        $\alpha_0 \leftarrow \det(\mathbf{M_{1,1}}(\mathbf{A_a}))$

        transfer function $G_j = \frac{\alpha_j}{\alpha_0}$      given in Equation (4)

        return $G_j$

    If Requirement == loop group gain (given node: $p_i$, and $p_k$)

        forward path= $F \leftarrow \det(^{(j)}H(p_1, \cdots, p_j)|_{(n,n)}^{(m,m+j)})$    given in Section 4

      loop group grain= $LG \leftarrow a_{k,\ i} \cdot F$

        return $LG$

---

## 4. Graph Decomposition

The recursive reduction of a signal flow graph proceeds by cofactor expansion. Each cofactor expansion reduces the order of the associated matrix by one. Layered (or hierarchical) graph drawing is a type of graph drawing in which the vertices of a directed graph are drawn in horizontal rows or layers with the edges generally directed downward. The number of feedback layers correlates to the number of strictly backward edges that form feedback loops during the reduction process. An alternative matrix form, as defined below, reveals the multiple feedback layers of the signal flow graph.

**Definition 1.** *An $(n - m - j + 1) \times (n - m - j + 1)$ square matrix $^{(j)}H(p_1, \cdots, p_j)|_{(n,n)}^{(m,m+j)}$ with multiple feedback layers $j$ is shown as.*

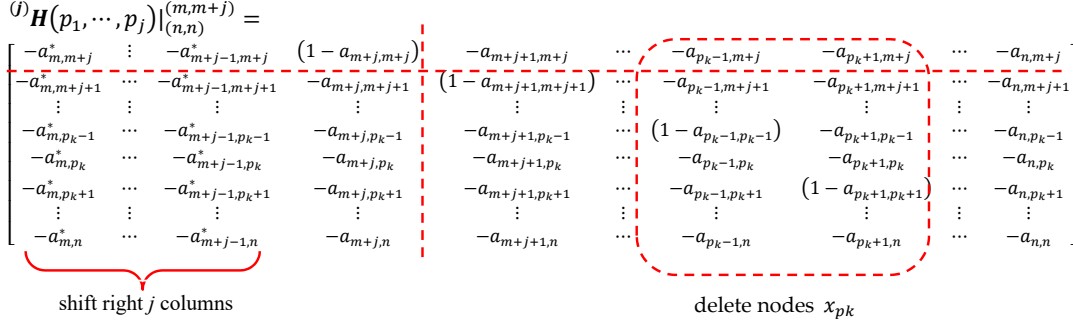

where $a_{i,j}$ and $a_{i,j}^*$ indicate the directed edge and equivalent weight of the forward edge from node $x_i$ to node $x_j$, respectively. When the edge is a self-loop edge, the value is $1 - a_{i,i}$; otherwise, the value is $-a_{i,j}$ or $-a_{i,j}^*$. The right upper index $(m, m+j)$ and lower index $(n, n)$ denote that the elements at the top entry $(1, 1)$ and bottom entry $(n - m - j + 1, n - m - j + 1)$ are a forward edge from node $x_m$ to node $x_{m+j}$ and the self-edge of the last node $x_n$, respectively. The arguments from $p_1$ to $p_j$ in descending order denote that the deleted columns are associated with the nodes from $x_{p_1}$ to $x_{p_j}$, respectively. The matrix should shift the coordinate of the characteristic element $(1 - a_{m+j,m+j})$ at the first row to the right by $j$ columns when the total number of deleted nodes is $j$. The value $j$ given in the left upper index is called the feedback layer.

**Example 1.** *Consider the signal flow diagram shown in Figure 2a. The matrix $\mathbf{M_{1,1}(A_a)}$ obtained by eliminating the first row and first column of the augmented matrix $\mathbf{A_a}$ is shown as*

$$\mathbf{M_{1,1}(A_a)} = \begin{bmatrix} 1 & 0 & 0 & 0 & -a_{5,1} & 0 \\ -a_{1,2} & 1 & 0 & 0 & 0 & -a_{6,2} \\ -a_{1,3} & -a_{2,3} & 1 & 0 & 0 & 0 \\ 0 & 0 & -a_{3,4} & 1 & 0 & 0 \\ 0 & -a_{2,5} & 0 & -a_{4,5} & 1 & 0 \\ 0 & 0 & 0 & -a_{4,6} & -a_{5,6} & 1 \end{bmatrix} \tag{5}$$

*The determinant of $\mathbf{M_{1,1}(A_a)}$ is expressed as*

$$\begin{aligned} \det(\mathbf{M_{1,1}(A_a)}) \quad &= 1 - a_{5,1}\left[(a_{1,2}a_{2,3} + a_{1,3})a_{3,4}a_{4,5} + a_{1,2}a_{2,5}\right] \\ &\quad - a_{6,2}\left[(a_{2,3}a_{3,4}a_{4,5} + a_{2,5})a_{5,6} + a_{2,3}a_{3,4}a_{4,6}\right] \\ &\quad - a_{6,2}a_{5,1}\left[a_{1,3}a_{3,4}a_{4,6}a_{2,5}\right] \end{aligned} \tag{6}$$

Figure 3 shows the intuitive approach to find each minor in the matrix of minors. The matrix of minors represents the signal flow diagram beside each cofactor calculations done in Figure 3. The loop group consists of the strictly backward edge and the forward path. This graph includes two strictly backward edges: $a_{5,1}$ and $a_{6,2}$. As shown in Figure 2b, the forward path gain that traverses paths from the input node $x_1$ to the node $x_5$ in the direction of the graph flow is $(a_{1,2}a_{2,3} + a_{1,3})a_{3,4}a_{4,5} + a_{1,2}a_{2,5}$. Similarly, the forward path gain from the input node $x_2$ to the node $x_6$ is $(a_{2,3}a_{3,4}a_{4,5} + a_{2,5})a_{5,6} + a_{2,3}a_{3,4}a_{4,6}$. The red solid and red dotted lines indicate the backward and forward paths, respectively. Figure 3, Part A calculates the determinant of cofactor matrix with respect to 1, and hence there is no signal flow through the nodes $a_{1,3}$, $a_{1,2}$ and a backward edge $a_{5,1}$. The later cofactor calculations have been done in accordance of calculating the whole determinant. In each figure, a cross marked line determines no signal flow. Figure 3, Part B calculates the determinant of cofactor matrix with respect to $a_{5,1}$ and hence no signal flows through $a_{5,6}$. The products of the loop gains with these two associated backward edges are as

$$\{LG_1\} = a_{5,1}\left[(a_{1,2}a_{2,3} + a_{1,3})a_{3,4}a_{4,5} + a_{1,2}a_{2,5}\right] \tag{7a}$$

$$\{LG_2\} = a_{6,2}\left[(a_{2,3}a_{3,4}a_{4,5} + a_{2,5})a_{5,6} + a_{2,3}a_{3,4}a_{4,6}\right] \tag{7b}$$

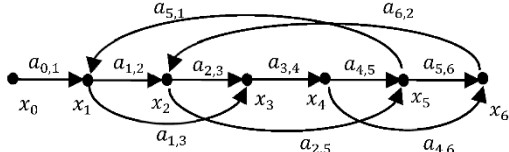

(**a**)  A signal flow graph.

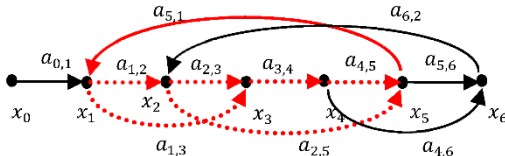

Loop group 1:  $\{LG_1\} = a_{5,1}[(a_{1,2}a_{2,3} + a_{1,3})a_{3,4}a_{4,5} + a_{1,2}a_{2,5}]$

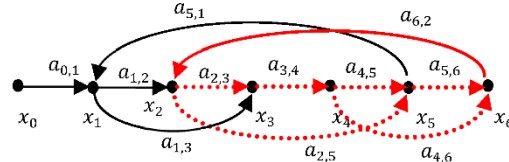

Loop group 2:  $\{LG_2\} = a_{6,2}\big[(a_{2,3}a_{3,4}a_{4,5} + a_{2,5})a_{5,6} + a_{2,3}a_{3,4}a_{4,6}\big]$

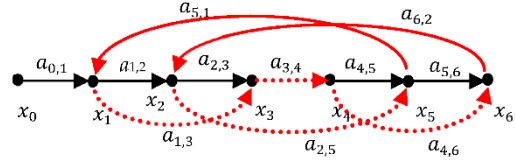

Contacting loop group (1 & 2):  $\{LG_1 \& LG_2\} = a_{6,2}a_{5,1}(a_{1,3}a_{3,4}a_{4,6}a_{2,5})$

(**b**) Loop groups.

**Figure 2.** A signal flow graph for example 1.

Although the loop group $\{LG_1\}$ overlaps loop group $\{LG_2\}$ at nodes $x_2$, $x_3$, $x_4$, and $x_5$, a forward path that starts and ends at node $x_1$ and passes through nodes $x_3$, $x_4$, $x_6$, $x_2$, and $x_5$ exists. The forward gain is $a_{1,3}a_{3,4}a_{4,6}a_{2,5}$. One touching loop combination exists of these two strictly backward edges with loop gains

$$\left\{ LG_1 \& LG_2|_{touching} \right\} = a_{6,2}a_{5,1}[a_{1,3}a_{3,4}a_{4,6}a_{2,5}] \tag{8}$$

The determinant of the graph may be expressed in Mason's gain formula by alternating the sign of the two touching loop groups.

$$\Delta = 1 - [\{LG_1\} + \{LG_2\}] + \left[(-1)\cdot\left\{ LG_1 \& LG_2|_{touching} \right\}\right] \tag{9}$$

The traditional Mason's gain formula includes seven simple loops with loop gains:

$$
\begin{aligned}
\{L_1\} &= a_{5,1}a_{1,2}a_{2,3}a_{3,4}a_{4,5} \\
\{L_2\} &= a_{5,1}a_{1,3}a_{3,4}a_{4,5} \\
\{L_3\} &= a_{5,1}a_{1,2}a_{2,5} \\
\{L_4\} &= a_{6,2}a_{2,3}a_{3,4}a_{4,5}a_{5,6} \\
\{L_5\} &= a_{6,2}a_{2,5}a_{5,6} \\
\{L_6\} &= a_{6,2}a_{2,3}a_{3,4}a_{4,6} \\
\{L_7\} &= a_{6,2}a_{5,1}a_{1,3}a_{3,4}a_{4,6}a_{2,5}
\end{aligned}
\tag{10}
$$

Two of these seven loops touch each other. Thus, the determinant of the graph yields

$$\Delta = 1 - \left[ \{L_1\} + \{L_2\} + \{L_3\} + \{L_4\} + \{L_5\} + \{L_6\} + \{L_7\} \right] \tag{11}$$

The loop group approach around the associated backward edges and traditional simple loop approach yield the same result.

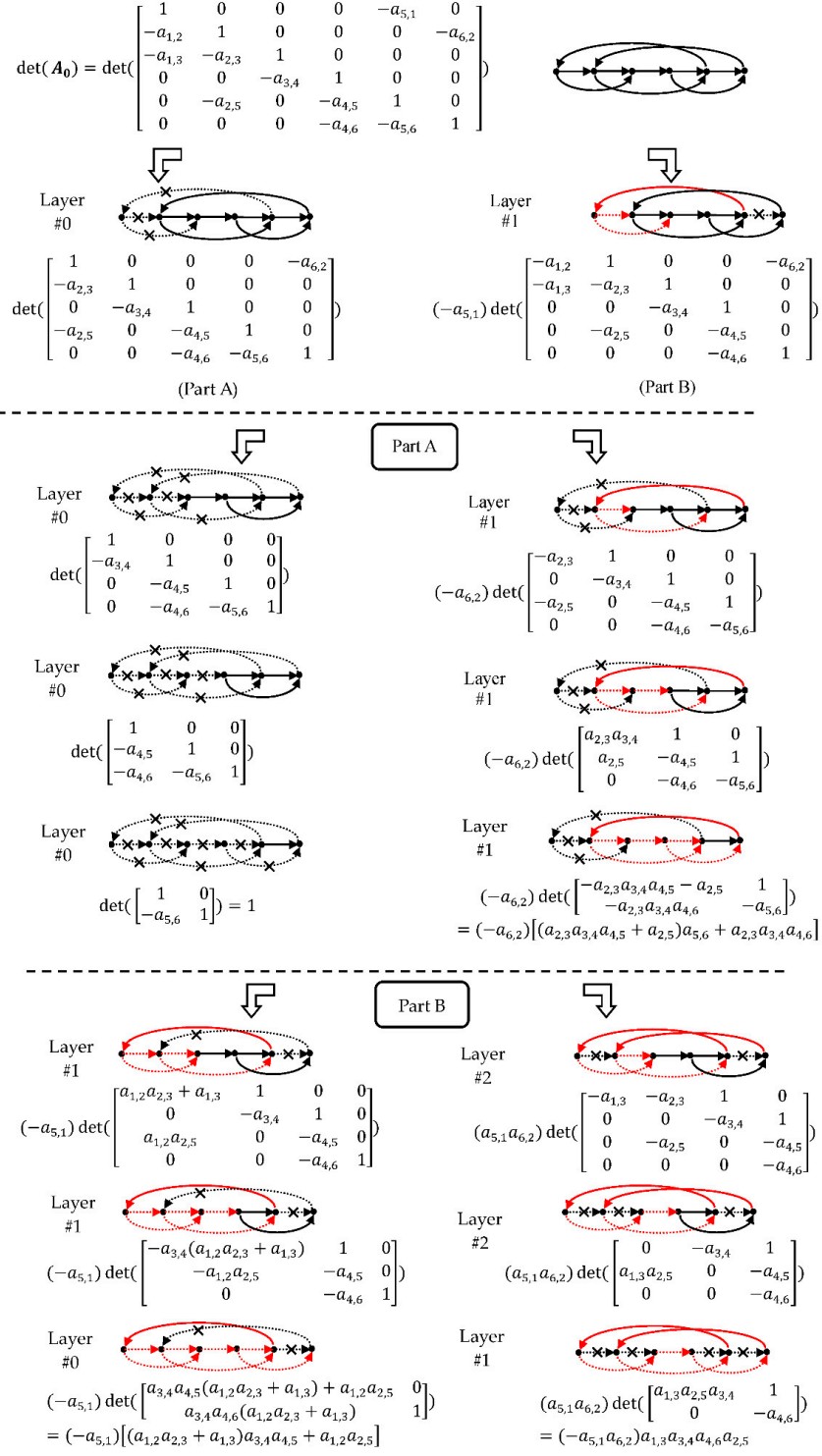

**Figure 3.** Graph decomposition of example 1.

**Example 2.** *Consider the signal flow diagram shown in Figure 4a. The matrix* $\mathbf{M_{1,1}(A_a)}$ *obtained by eliminating the first row and first column of the augmented matrix* $\mathbf{A_a}$ *is*

$$
\mathbf{M_{1,1}(A_a)} = \begin{bmatrix}
1 & 0 & 0 & 0 & 0 & 0 & 0 & 0 & -a_{9,1} \\
-a_{1,2} & 1 & 0 & 0 & 0 & 0 & 0 & 0 & 0 \\
0 & -a_{2,3} & 1 & -a_{4,3} & 0 & 0 & 0 & 0 & 0 \\
0 & 0 & -a_{3,4} & 1 & -a_{5,4} & 0 & 0 & 0 & 0 \\
0 & 0 & 0 & -a_{4,5} & 1 & 0 & 0 & 0 & 0 \\
0 & -a_{2,6} & 0 & 0 & 0 & 1 & -a_{7,6} & 0 & 0 \\
0 & 0 & 0 & 0 & 0 & -a_{6,7} & 1 & -a_{8,7} & 0 \\
0 & 0 & 0 & 0 & 0 & 0 & -a_{7,8} & 1 & 0 \\
0 & 0 & 0 & 0 & -a_{5,9} & 0 & 0 & -a_{8,9} & 1
\end{bmatrix} \tag{12}
$$

*Using the cofactor expansion of the determinant along the first row iteratively yields*

$$
\det(\mathbf{M_{1,1}(A_a)}) = (1 - a_{4,3}a_{3,4} - a_{5,4}a_{4,5})(1 - a_{7,6}a_{6,7} - a_{8,7}a_{7,8})
$$
$$
- a_{9,1}\begin{bmatrix} a_{1,2}a_{2,3}a_{3,4}a_{4,5}a_{5,9}(1 - a_{6,7}a_{7,6} - a_{7,8}a_{8,7}) \\ +a_{1,2}a_{2,6}a_{6,7}a_{7,8}a_{8,9}(1 - a_{4,3}a_{3,4} - a_{5,4}a_{4,5}) \end{bmatrix} \tag{13}
$$

*Figure 4b shows five strictly backward edges:* $a_{4,3}$, $a_{5,4}$, $a_{7,6}$, $a_{8,7}$, *and* $a_{9,1}$ *in the signal flow graph. The forward path gains for the first four backward edges are* $a_{3,4}$, $a_{4,5}$, $a_{6,7}$, *and* $a_{7,8}$, *respectively. Supposing that* $^{(1)}\mathbf{H}(9)|_{(8,9)}^{(1,2)}$ *be a matrix of one feedback layer, which results from the elimination of the first row and the last column of the matrix* $\mathbf{M_{1,1}(A_a)}$, *the mathematical expression of* $^{(1)}\mathbf{H}(9)|_{(8,9)}^{(1,2)}$ *is shown as*

$$
^{(1)}\mathbf{H}(9)|_{(9,9)}^{(1,2)} = \begin{bmatrix}
-a_{1,2} & 1 & 0 & 0 & 0 & 0 & 0 & 0 \\
0 & -a_{2,3} & 1 & -a_{4,3} & 0 & 0 & 0 & 0 \\
0 & 0 & -a_{3,4} & 1 & -a_{5,4} & 0 & 0 & 0 \\
0 & 0 & 0 & -a_{4,5} & 1 & 0 & 0 & 0 \\
0 & -a_{2,6} & 0 & 0 & 0 & 1 & -a_{7,6} & 0 \\
0 & 0 & 0 & 0 & 0 & -a_{6,7} & 1 & -a_{8,7} \\
0 & 0 & 0 & 0 & 0 & 0 & -a_{7,8} & 1 \\
0 & 0 & 0 & 0 & -a_{5,9} & 0 & 0 & -a_{8,9}
\end{bmatrix} \tag{14}
$$

*The determinant of* $^{(1)}\mathbf{H}(9)|_{(9,9)}^{(1,2)}$ *is the gain for the possible forward paths from node* $x_1$ *to node* $x_9$. *The strictly backward edges result in five loop groups with loop group gains.*

$$
\{LG_1\} = a_{4,3}a_{3,4}
$$
$$
\{LG_2\} = a_{5,4}a_{4,5}
$$
$$
\{LG_3\} = a_{7,6}a_{6,7}
$$
$$
\{LG_4\} = a_{8,7}a_{7,8} \tag{15}
$$
$$
\{LG_5\} = a_{9,1}\det(^{(1)}\mathbf{H}(9)|_{(9,9)}^{(1,2)})
$$
$$
= a_{9,1}[a_{1,2}a_{2,3}a_{3,4}a_{4,5}a_{5,9}(1 - a_{6,7}a_{7,6} - a_{7,8}a_{8,7}) +
$$
$$
a_{1,2}a_{2,6}a_{6,7}a_{7,8}a_{8,9}(1 - a_{4,3}a_{3,4} - a_{5,4}a_{4,5})]
$$

*The non-touching loop groups are identified two at a time. Loop group 1 does not touch loop groups 3 or 4, and neither does loop group 2. Notably, loop groups 1–4 all touch loop group 5. Thus, the combinations of the non-touching loop gains identified two at a time are obtained as*

$$\begin{cases} LG_1 \& LG_3|_{non-touching} \} = \{LG_1\} \cdot \{LG_3\} = (a_{4,3}a_{3,4})(a_{7,6}a_{6,7}) \\ LG_1 \& LG_4|_{non-touching} \} = \{LG_1\} \cdot \{LG_4\} = (a_{4,3}a_{3,4})(a_{8,7}a_{7,8}) \\ LG_2 \& LG_3|_{non-touching} \} = \{LG_2\} \cdot \{LG_3\} = (a_{5,4}a_{4,5})(a_{7,6}a_{6,7}) \\ LG_2 \& LG_4|_{non-touching} \} = \{LG_2\} \cdot \{LG_4\} = (a_{5,4}a_{4,5})(a_{8,7}a_{7,8}) \end{cases} \tag{16}$$

*No combination exists of more than two non-touching loop groups. According to Mason's gain formula, the determinant of the graph may be rewritten as*

$$\det(\mathbf{M_{1,1}}(\mathbf{A_a})) = [1 - \{LG_1\} - \{LG_2\}] \cdot [1 - \{LG_3\} - \{LG_4\}] - \{LG_5\} \tag{17}$$

*which is identical to Equation (13). Figure 5 shows the approach to calculate the minors at each decomposition.*

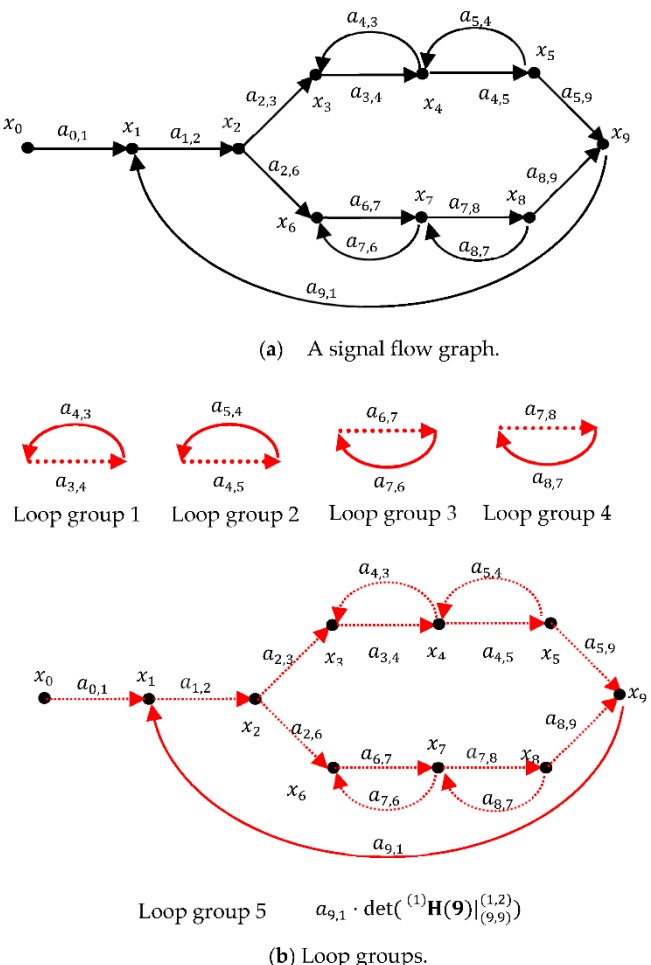

(**a**)　A signal flow graph.

Loop group 1　　Loop group 2　　Loop group 3　　Loop group 4

Loop group 5　　　$a_{9,1} \cdot \det(\,^{(1)}\mathbf{H(9)}|_{(9,9)}^{(1,2)})$

(**b**) Loop groups.

**Figure 4.** A signal flow graph for example 2.

$$\det(\mathbf{A_0}) = \det\left(\begin{bmatrix} 1 & 0 & 0 & 0 & 0 & 0 & 0 & 0 & -a_{9,1} \\ -a_{1,2} & 1 & 0 & 0 & 0 & 0 & 0 & 0 & 0 \\ 0 & -a_{2,3} & 1 & -a_{4,3} & 0 & 0 & 0 & 0 & 0 \\ 0 & 0 & -a_{3,4} & 1 & -a_{5,4} & 0 & 0 & 0 & 0 \\ 0 & 0 & 0 & -a_{4,5} & 1 & 0 & 0 & 0 & 0 \\ 0 & -a_{2,6} & 0 & 0 & 0 & 1 & -a_{7,6} & 0 & 0 \\ 0 & 0 & 0 & 0 & 0 & -a_{6,7} & 1 & -a_{8,7} & 0 \\ 0 & 0 & 0 & 0 & 0 & 0 & -a_{7,8} & 1 & 0 \\ 0 & 0 & 0 & 0 & -a_{5,9} & 0 & 0 & -a_{8,9} & 1 \end{bmatrix}\right)$$

$$\det\left(\begin{bmatrix} 1 & 0 & 0 & 0 & 0 & 0 & 0 & 0 \\ -a_{2,3} & 1 & -a_{4,3} & 0 & 0 & 0 & 0 & 0 \\ 0 & -a_{3,4} & 1 & -a_{5,4} & 0 & 0 & 0 & 0 \\ 0 & 0 & -a_{4,5} & 1 & 0 & 0 & 0 & 0 \\ -a_{2,6} & 0 & 0 & 0 & 1 & -a_{7,6} & 0 & 0 \\ 0 & 0 & 0 & 0 & -a_{6,7} & 1 & -a_{8,7} & 0 \\ 0 & 0 & 0 & 0 & 0 & -a_{7,8} & 1 & 0 \\ 0 & 0 & 0 & -a_{5,9} & 0 & 0 & -a_{8,9} & 1 \end{bmatrix}\right)$$

$$(-a_{9,1})\det\left(\begin{bmatrix} -a_{1,2} & 1 & 0 & 0 & 0 & 0 & 0 & 0 \\ 0 & -a_{2,3} & 1 & -a_{4,3} & 0 & 0 & 0 & 0 \\ 0 & 0 & -a_{3,4} & 1 & -a_{5,4} & 0 & 0 & 0 \\ 0 & 0 & 0 & -a_{4,5} & 1 & 0 & 0 & 0 \\ 0 & -a_{2,6} & 0 & 0 & 0 & 1 & -a_{7,6} & 0 \\ 0 & 0 & 0 & 0 & 0 & -a_{6,7} & 1 & -a_{8,7} \\ 0 & 0 & 0 & 0 & 0 & 0 & -a_{7,8} & 1 \\ 0 & 0 & 0 & 0 & -a_{5,9} & 0 & 0 & -a_{8,9} \end{bmatrix}\right)$$

$$\det\left(\begin{bmatrix} 1 & -a_{4,3} & 0 & 0 & 0 & 0 & 0 \\ -a_{3,4} & 1 & -a_{5,4} & 0 & 0 & 0 & 0 \\ 0 & -a_{4,5} & 1 & 0 & 0 & 0 & 0 \\ 0 & 0 & 0 & 1 & -a_{7,6} & 0 & 0 \\ 0 & 0 & 0 & -a_{6,7} & 1 & -a_{8,7} & 0 \\ 0 & 0 & 0 & 0 & -a_{7,8} & 1 & 0 \\ 0 & 0 & -a_{5,9} & 0 & 0 & -a_{8,9} & 1 \end{bmatrix}\right)$$

$$(-a_{9,1})\det\left(\begin{bmatrix} a_{1,2}a_{2,3} & 1 & -a_{4,3} & 0 & 0 & 0 & 0 \\ 0 & -a_{3,4} & 1 & -a_{5,4} & 0 & 0 & 0 \\ 0 & 0 & -a_{4,5} & 1 & 0 & 0 & 0 \\ a_{1,2}a_{2,6} & 0 & 0 & 0 & 1 & -a_{7,6} & 0 \\ 0 & 0 & 0 & 0 & -a_{6,7} & 1 & -a_{8,7} \\ 0 & 0 & 0 & 0 & 0 & -a_{7,8} & 1 \\ 0 & 0 & 0 & -a_{5,9} & 0 & 0 & -a_{8,9} \end{bmatrix}\right)$$

$$\det\left(\begin{bmatrix} 1 & -a_{5,4} & 0 & 0 & 0 & 0 \\ -a_{4,5} & 1 & 0 & 0 & 0 & 0 \\ 0 & 0 & 1 & -a_{7,6} & 0 & 0 \\ 0 & 0 & -a_{6,7} & 1 & -a_{8,7} & 0 \\ 0 & 0 & 0 & -a_{7,8} & 1 & 0 \\ 0 & -a_{5,9} & 0 & 0 & -a_{8,9} & 1 \end{bmatrix}\right)$$

$$(-a_{9,1})\det\left(\begin{bmatrix} -a_{1,2}a_{2,3}a_{3,4} & 1 & -a_{5,4} & 0 & 0 & 0 \\ 0 & -a_{4,5} & 1 & 0 & 0 & 0 \\ -a_{1,2}a_{2,6} & 0 & 0 & 1 & -a_{7,6} & 0 \\ 0 & 0 & 0 & -a_{6,7} & 1 & -a_{8,7} \\ 0 & 0 & 0 & 0 & -a_{7,8} & 1 \\ 0 & 0 & -a_{5,9} & 0 & 0 & -a_{8,9} \end{bmatrix}\right)$$

$$(a_{4,3})\det\left(\begin{bmatrix} -a_{3,4} & -a_{5,4} & 0 & 0 & 0 & 0 \\ 0 & 1 & 0 & 0 & 0 & 0 \\ 0 & 0 & 1 & -a_{7,6} & 0 & 0 \\ 0 & 0 & -a_{6,7} & 1 & -a_{8,7} & 0 \\ 0 & 0 & 0 & -a_{7,8} & 1 & 0 \\ 0 & -a_{5,9} & 0 & 0 & -a_{8,9} & 1 \end{bmatrix}\right)$$

$$(a_{9,1}a_{4,3})\det\left(\begin{bmatrix} 0 & -a_{3,4} & -a_{5,4} & 0 & 0 & 0 \\ 0 & 0 & 1 & 0 & 0 & 0 \\ a_{1,2}a_{2,6} & 0 & 0 & 1 & -a_{7,6} & 0 \\ 0 & 0 & 0 & -a_{6,7} & 1 & -a_{8,7} \\ 0 & 0 & 0 & 0 & -a_{7,8} & 1 \\ 0 & 0 & -a_{5,9} & 0 & 0 & -a_{8,9} \end{bmatrix}\right)$$

**Figure 5.** *Cont.*

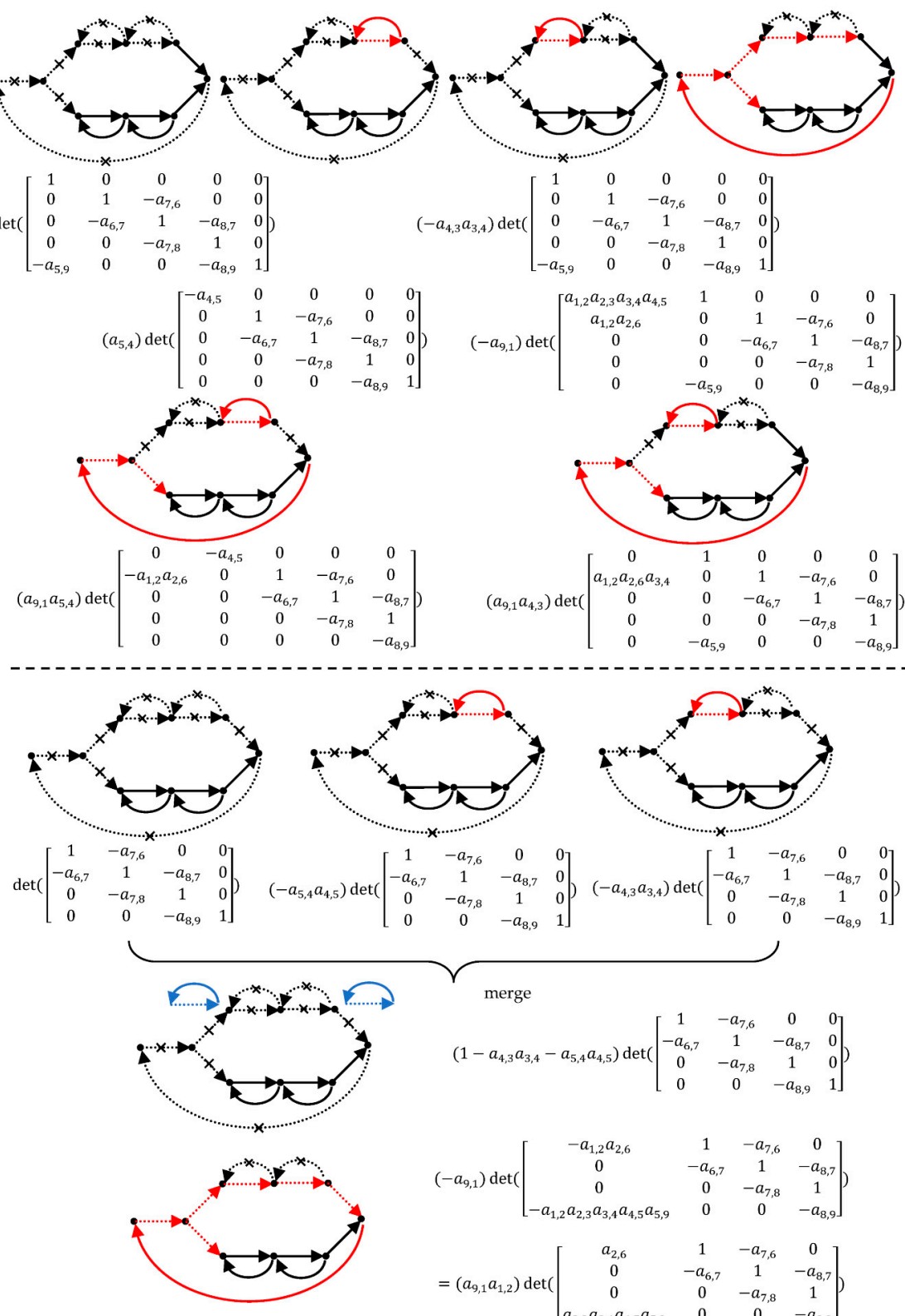

**Figure 5.** *Cont.*

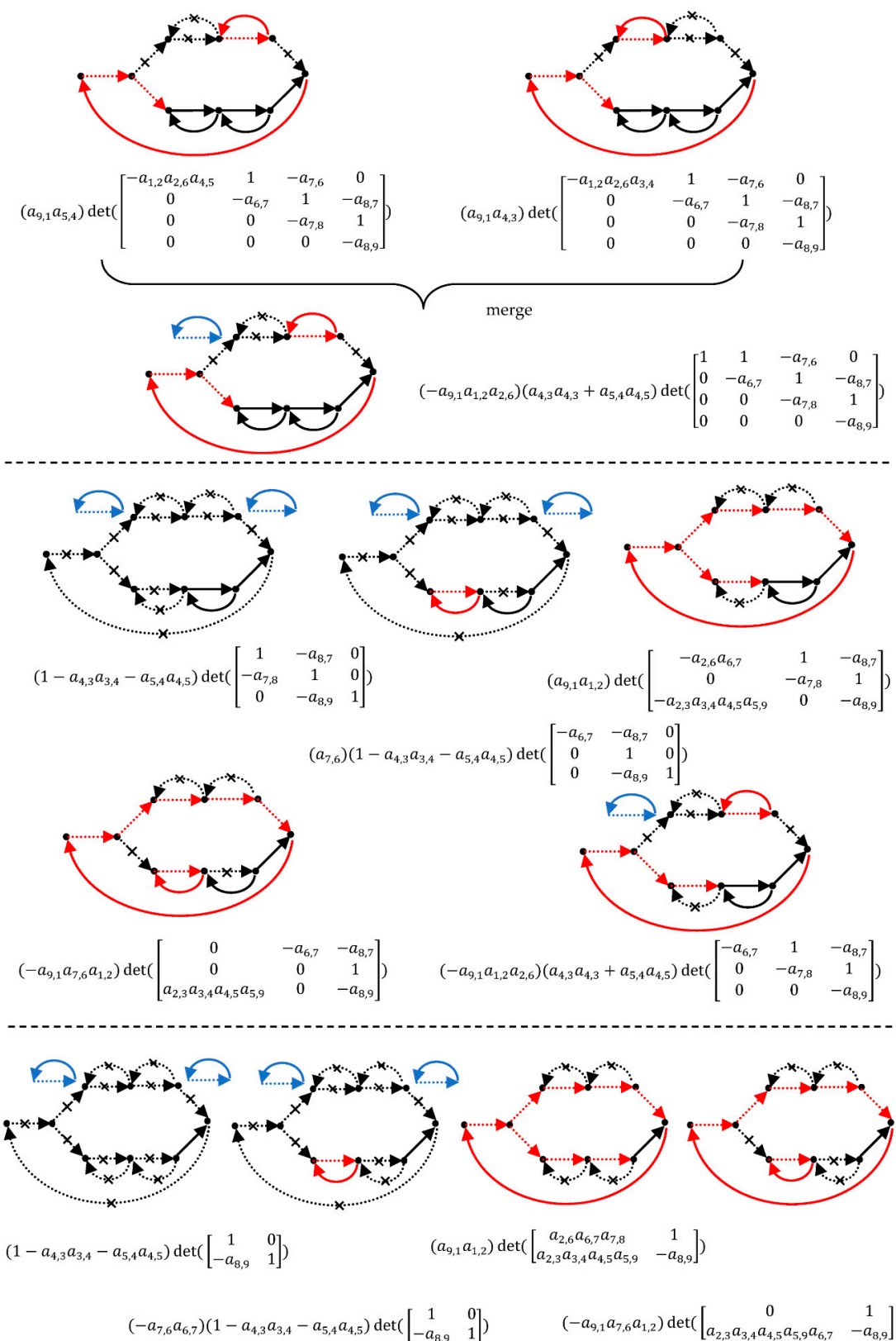

**Figure 5.** *Cont.*

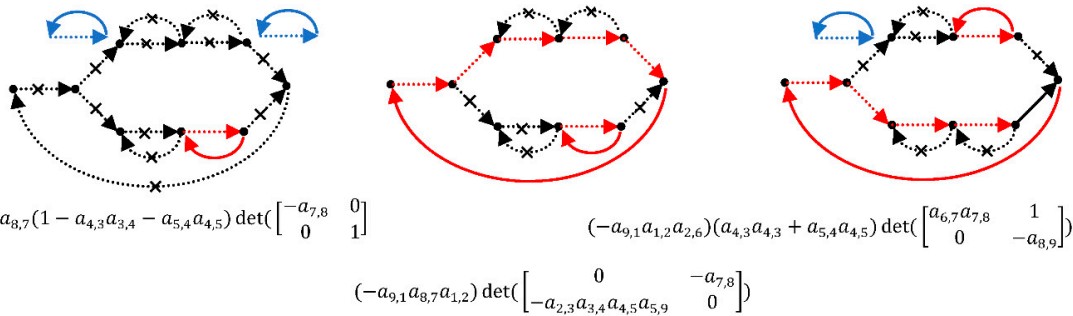

$$a_{8,7}(1 - a_{4,3}a_{3,4} - a_{5,4}a_{4,5}) \det\left(\begin{bmatrix} -a_{7,8} & 0 \\ 0 & 1 \end{bmatrix}\right) \qquad (-a_{9,1}a_{1,2}a_{2,6})(a_{4,3}a_{4,3} + a_{5,4}a_{4,5}) \det\left(\begin{bmatrix} a_{6,7}a_{7,8} & 1 \\ 0 & -a_{8,9} \end{bmatrix}\right)$$

$$(-a_{9,1}a_{8,7}a_{1,2}) \det\left(\begin{bmatrix} 0 & -a_{7,8} \\ -a_{2,3}a_{3,4}a_{4,5}a_{5,9} & 0 \end{bmatrix}\right)$$

**Figure 5.** Graph decomposition of example 2.

## 5. Forward Path Gain

The explanations of Figure 5 are similar to Figure 3 which explains the signal flow visualization while calculating the determinant of each cofactor. From Figures 2–5, the red dashed lines and red solid lines indicate the possible forward paths and the associated backward edge, respectively. Consider the strictly backward edge $a_{5,1}$ of Example 1 shown in Figure 2. The matrix $^{(1)}\mathbf{H}(5)|_{(6,6)}^{(1,2)}$ resulting from the elimination of the incoming branches of the sink node $x_1$ and outgoing branches of the source node $x_5$ with one feedback layer is.

$$^{(1)}\mathbf{H}(5)|_{(6,6)}^{(1,2)} = \begin{bmatrix} -a_{1,2} & 1 & 0 & 0 & -a_{6,2} \\ -a_{1,3} & -a_{2,3} & 1 & 0 & 0 \\ 0 & 0 & -a_{3,4} & 1 & 0 \\ 0 & -a_{2,5} & 0 & -a_{4,5} & 0 \\ 0 & 0 & 0 & -a_{4,6} & 1 \end{bmatrix} \tag{18}$$

The determinant of $^{(1)}\mathbf{H}(5)|_{(6,6)}^{(1,2)}$ is $[(a_{1,2}a_{2,3} + a_{1,3})a_{3,4}a_{4,5} + a_{1,2}a_{2,5}] + [a_{6,2}a_{1,3}a_{3,4}a_{4,6}a_{2,5}]$. The first term is identical to the forward path gain of $\{LG_1\}$, and the second term equals the product of the strictly backward edge $a_{6,2}$ and the forward path gain of $\{LG_1 \& LG_2|_{touching}\}$. These two terms reveal the total forward gain from node $x_1$ to node $x_5$. The matrix $^{(2)}\mathbf{H}(5,6)|_{(6,6)}^{(1,3)}$ obtained by deleting the source and sink nodes of two strictly backward edges $a_{5,1}$ and $a_{6,2}$ with two feedback layers can be expressed as

$$^{(2)}\mathbf{H}(5,6)|_{(6,6)}^{(1,3)} = \begin{bmatrix} -a_{1,3} & -a_{2,3} & 1 & 0 \\ 0 & 0 & -a_{3,4} & 1 \\ 0 & -a_{2,5} & 0 & -a_{4,5} \\ 0 & 0 & 0 & -a_{4,6} \end{bmatrix} \tag{19}$$

The determinant of $^{(2)}\mathbf{H}(5,6)|_{(6,6)}^{(1,3)}$ is $a_{1,3}a_{3,4}a_{4,6}a_{2,5}$; this equals the forward path gain of $\{LG_1 \& LG_2|_{touching}\}$. Figure 3 shows how some of the forward path gain associated with the other strictly backward edge $a_{6,2}$ is verified.

As shown in Figure 4b, consider Mason's gain formula of the possible forward paths with the associated backward edge $a_{9,1}$ for Example 2, which involves two forward paths with path gains.

$$P_1 = a_{1,2}a_{2,3}a_{3,4}a_{4,5}a_{5,9} \qquad P_2 = a_{1,2}a_{2,6}a_{6,7}a_{7,8}a_{8,9} \tag{20}$$

Considering $P_1$, neither loop group 3 nor 4 touches the first forward path. According to Mason's gain formula, the cofactor along the first forward path $P_1$, when the loops touching the first forward path are eliminated, is $\Delta_1 = 1 - \{LG_3\} - \{LG_4\}$. For $P_2$, neither loop group 1 nor 2 touches the second

forward path. The cofactor along the second forward path $P_2$ is $\Delta_2 = 1 - \{LG_1\} - \{LG_2\}$. The total forward path gain from node $x_1$ to node $x_9$ is

$$\sum_{k=1}^{2} P_k \Delta_k = a_{1,2}a_{2,3}a_{3,4}a_{4,5}a_{5,9}(1 - a_{6,7}a_{7,6} - a_{7,8}a_{8,7}) + a_{1,2}a_{2,6}a_{6,7}a_{7,8}a_{8,9}(1 - a_{4,3}a_{3,4} - a_{5,4}a_{4,5}) \qquad (21)$$

which is the same result as that calculated by the determinant of $^{(1)}\mathbf{H}(9)|_{(9,9)}^{(1,2)}$ in Equation (14). The matrix $^{(1)}\mathbf{H}(9)|_{(9,9)}^{(1,2)}$ is formed by eliminating the first row and the ninth column of matrix $\mathbf{M_{1,1}}(\mathbf{A_a})$, which removes the incoming branches of input node $x_1$ and outgoing branches of backward node $x_9$. The product term $(a_{9,1}) \cdot \det(^{(1)}\mathbf{H}(9)|_{(8,9)}^{(1,2)})$ is the loop group gain $\{LG_5\}$ with the strictly backward edge $a_{9,1}$.

## 6. Transfer Function

In Mason's gain formula, the transfer function is expressed as a ratio of the numerator terms to the denominator terms. The numerator and denominator terms are located in the forward paths and feedback loops, respectively. According to Equation (4), the numerator term of the transfer function for Example 2 can be obtained by determining the cofactor matrix of the augmented matrix $\mathbf{A_a}$, at entry (1, 10).

$$-\mathbf{M_{1,10}}(\mathbf{A_a})$$

$$= (-1) \begin{bmatrix} -a_{0,1} & 1 & 0 & 0 & 0 & 0 & 0 & 0 & 0 \\ 0 & -a_{1,2} & 1 & 0 & 0 & 0 & 0 & 0 & 0 \\ 0 & 0 & -a_{2,3} & 1 & -a_{4,3} & 0 & 0 & 0 & 0 \\ 0 & 0 & 0 & -a_{3,4} & 1 & -a_{5,4} & 0 & 0 & 0 \\ 0 & 0 & 0 & 0 & -a_{4,5} & 1 & 0 & 0 & 0 \\ 0 & 0 & -a_{2,6} & 0 & 0 & 0 & 1 & -a_{7,6} & 0 \\ 0 & 0 & 0 & 0 & 0 & 0 & -a_{6,7} & 1 & -a_{8,7} \\ 0 & 0 & 0 & 0 & 0 & 0 & 0 & -a_{7,8} & 1 \\ 0 & 0 & 0 & 0 & 0 & -a_{5,9} & 0 & 0 & -a_{8,9} \end{bmatrix} \qquad (22)$$

Figure 6 shows the original signal flow diagram by adding a virtual backward edge $a_{9,0}$ from output node $x_9$ to input node $x_0$. The determinant of the matrix $-\mathbf{M_{1,10}}(\mathbf{A_a})$ having one feedback layer for the coordinate of the characteristic element shifting to the right by one column at the first row is equivalent to directly obtaining the forward gain with the virtual backward edge $a_{9,0}$. The forward gain is

$$G_9 = \frac{\det(-\mathbf{M_{1,10}}(\mathbf{A_a}))}{\det(\mathbf{M_{1,1}}(\mathbf{A_a}))} = \frac{a_{0,1} \cdot \{LG_5\}}{[1 - \{LG_1\} - \{LG_2\}] \cdot [1 - \{LG_3\} - \{LG_4\}] - \{LG_5\}} \qquad (23)$$

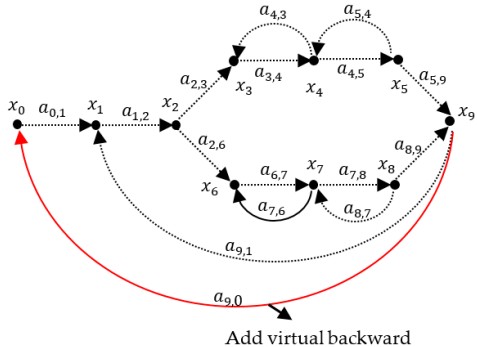

**Figure 6.** The forward path gain of the virtual loop group around the entire system for example 2.

## 7. Complexity Analysis

Mason's rule can increase factorially because the enumeration of paths in a directed graph grows dramatically. Consider a complete directed graph on $n$ vertices, with an edge between every pair of vertices. The paths from $x_0$ to $x_n$ are $(n-2)!$ permutations of the intermediate vertices. Mason's graph formula describes the transfer function of an interconnected system, which is divided into several independent loops and forward paths, and simultaneously performs algebraic and combinatorial operation. According to Equation (4), the transfer function may be calculated by using the determinant of an $n \times n$ matrix. Given an $n \times n$ determinant to be calculated, we can either use the cofactor method recursively with a runtime of O($n!$), or use Gaussian elimination method to simplify the matrix, track the influence on the determinant, and then multiply it by the diagonal entries at the end. This will be O($n^3$), the order of Gaussian elimination.

## 8. Conclusions

The loop group approach presented in this paper yields the same results as Mason's gain formula. In addition, we reduced the signal flow graph recursively through cofactor expansion to systematically obtain all possible non-touching loop combinations. A set of loops were used to share the same backward edges, referred to as a loop group, can simplify the loop enumeration. Each cofactor expansion reduces the order of the augmented matrix by one. The determinant of the cofactor matrix is equivalent to the loop group gain associated with the backward edge. It can also provide a physical insight of the signal flow.

An augmented matrix is used to represent the signal flow graph. The determinant of the cofactor matrix of the augmented matrix at the first entry containing no feedback layer is the denominator term of the transfer function. The numerator term that directs the forward path gain from the input node to the output node can be obtained by determining the cofactor matrix of the augmented matrix at the top-right entry. The matrix contains one feedback layer, and the determinant is the associated forward path gain of the outer virtual loop group. Two examples of feedback networks are used to demonstrate the intuitive approach to obtain the transfer function for both numerical and computer-aided symbolic analysis. The transfer matrix offers an excellent physical insight because it enables visualization of the signal flow.

**Author Contributions:** Conceptualization, S.-L.J.; methodology, W.-H.C.; data processing, R.R.; results analysis, S.-L.J.; Visualization, S.-L.J.; writing—original draft preparation, R.R.; writing—review and editing, R.R.; supervision, W.-H.C. All authors have read and agreed to the published version of the manuscript.

**Funding:** This research was funded by National Science Council of Taiwan, the Republic of China grant number MOST 109-2622-E-262-005 -CC3.

**Conflicts of Interest:** The authors declare no conflict of interest.

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
