# Peer review of "A Matrix Approach for Analyzing Signal Flow Graph"

_information, doi:10.3390/info11120562_

Round 1
Reviewer 1 Report
The manuscript entitled: "A Matrix Approach for Analyzing Signal Flow Graph” contributes in calculating the transfer function of a system.
Introduction is relatively short to describe authors’ research. Bibliography could be expanded. Authors’ contribution should be compared with all available alternatives. Which are the advantages and the disadvantages of the proposed method?
The methodological foundation in terms of algorithmic representation is not clear to the reviewer. The authors could provide a pseudocode of their approach to improve readability.
Conclusions are generic. It is necessary to deep into the importance of the method and explain to the readers why they should be using it.
Author Response
We as the authors express our appreciation to Reviewer 1 and the editor. We believe that their vital comments and suggestion have contributed substantially to improve the presentation of our study, as well as its overall quality of the manuscript. Following, we offer pointwise replies to the issues, the reviewers addressed regarding the original manuscript.

Reviewer 2 Report
This paper studies a matrix-based representation for signal flow graph. It is currently with some drawbacks mainly associated with presentation and evaluation. Please refer to the followings for the details.
- Two detailed examples are presented well. However, it is difficult for readers to reproduce the results for different graphs unless the proposed method is clearly summarized. There should be a rigorous description or at least a pseudocode for the method, and the two examples should be accordant with it.
- In Abstract, it is claimed that the proposed method is of lower complexity than the existing counterpart. However, no rigorous complexity analysis and comparison with the existing methods are available.
- Section 1 does not well state the drawbacks of the prior works. In other words, the paper is not motivated enough.
- It would be better to introduce Mason’s rule and preceding works at least in a brief manner.
- Is (1) correct? In its current form, A^T should be an identity matrix.
- Variables should be italicized properly, but some of those in page 3 are not.
- All the references are old. Please review the recent works in the literature.
Author Response
We as the authors express our appreciation to Reviewer 2 and the editor. We believe that their vital comments and suggestion have contributed and substantially to improve the presentation of our study, as well as its overall quality of the manuscript. Following, we offer pointwise replies to the issues, the reviewers addressed regarding the original manuscript.

Reviewer 3 Report
The authors present a method for analyzing the loop gain of a signal flow graph based on the transform matrix approach. They demonstrate the approach on two examples of feedback networks. Thre proposed method cann offer physical insights due to the visualization of the signal flow. The paper is well-structured and the method is presented in a concise way. I have only a couple of comments:
1. The incentive for this paper is presented in a single paragraph of Section 1. I believe that other relevant papers in the field should be also mentioned. Moreover, authors should state in Section 1 the advantages of their proposed method.
2. A summary of the organization of the paper is missing at the end of Section 1.
3. Some characters are off in a few Equations, e.g., the letter "h" in the word "touching" in Equation 16.
4. The description of Figures 2 - 5 need to be more detailed.
Author Response
We as the authors express our appreciation to Reviewer 3 and the editor. We believe that their vital comments and suggestion have contributed and substantially to improve the presentation of our study, as well as its overall quality of the manuscript. Following, we offer pointwise replies to the issues, the reviewers addressed regarding the original manuscript.

Round 2
Reviewer 2 Report
All the concerns of the reviewer have been satisfactorily addressed. Thank you for the authors' efforts.